# Multi-Mechanistic and Therapeutic Exploration of Nephroprotective Effect of Traditional Ayurvedic Polyherbal Formulation Using In Silico, In Vitro and In Vivo Approaches

**DOI:** 10.3390/biomedicines11010168

**Published:** 2023-01-09

**Authors:** Ikshit Sharma, Mohammad Umar Khan, Sultan Zahiruddin, Parakh Basist, Sayeed Ahmad

**Affiliations:** 1Centre of Excellence (CoE) in Unani Medicine (Pharmacognosy and Pharmacology), Bioactive Natural Product Laboratory, Jamia Hamdard, New Delhi 110062, India; 2AIMIL Pharmaceuticals (India) Ltd., Saini Majra, Ropar Nalagarh Road, Solan District, Nalagarh 174101, India

**Keywords:** NEERI KFT, nephrotoxicity, cisplatin, network pharmacology, antioxidant activity, anti-inflammatory activity

## Abstract

Based on traditional therapeutic claims, NEERI KFT (a traditional Ayurvedic polyherbal preparation) has been innovatively developed in recent time on the decades of experience for treating kidney dysfunction. Due to the lack of scientific evidence, the present investigations are needed to support the rationale use of NEERI KFT. Considering the facts, the study investigated the nephroprotective effect of NEERI KFT against kidney dysfunction using in silico, in vitro and in vivo approaches. In this study, phytochemical and network pharmacology studies were performed for the developed formulation to evaluate the molecular mechanism of NEERI KFT in the amelioration of kidney disease. In vitro nephroprotective and antioxidant effect of NEERI KFT was determined on HEK 293 cells against cisplatin-induced cytotoxicity and oxidative stress. In vivo nephroprotective effect of NEERI KFT was determined against cisplatin-induced nephrotoxicity in Wistar rats, via assessing biochemical markers, antioxidant enzymes and inflammatory cytokines such as TNF-α, IL-1β, CASP-3, etc. The results showed that the compounds such as gallic acid, caffeic acid and ferulic acid are the major constituents of NEERI KFT, while network pharmacology analysis indicated a strong interaction between polyphenols and several genes (CASPs, ILs, AGTR1, AKT, ACE2, SOD1, etc.) involved in the pathophysiology of kidney disease. In vivo studies showed a significant (*p* < 0.05) ameliorative effect on biochemical markers and antioxidant enzymes (SOD, CAT, GSH, etc.), and regulates inflammatory cytokine (TNF-α, IL-1β, CASP-3) expression in kidney tissue. Hence, it can be concluded that NEERI KFT subsequently alleviates renal dysfunction mediated by cisplatin via attenuating oxidative and inflammatory stress, thus preserving the normalcy of kidney function.

## 1. Introduction

The kidneys are mainly characterized as the main emunctory organ and facilitated to play essential physiological functions such as infiltration, toxin waste and metabolites excretion; internal fluid environment regulation to maintain proper tonicity and fluid volume; pH balance; electrolyte homeostasis; and essential endocrine functions [1,2]. As the main emunctory organ, the kidneys are generally subacted by the aggression of many toxins [3]. Cisplatin is considered a potent inorganic platinum-based oncologic medication practiced as a chemotherapy agent. Recent epidemiological studies have proposed that cisplatin is hyphenated with vigorous changes in the structural and functional unit of the kidneys, leading to acute and chronic renal failure [4]. However, cisplatin utilization is limited due to severe tissue damage, such as bone marrow suppression, ototoxicity, nephrotoxicity and neurotoxicity [5]. Cisplatin therapy, however, is still targeted to effectively reduce tumor burden and shows a narrow and unfavorable therapeutic index. Moreover, 70% of patients undergoing cisplatin chemotherapy are associated with signs of nephrotoxicity. Because of its high impact to risk ratio, cisplatin is still an integral part of the therapeutic combination for the treatment of various cancers [6]. To prevent the toxic effect of cisplatin on the kidney, there is a need for a preventive measure that significantly ameliorates cisplatin toxicity without compromising its efficacy. At present, there is a paucity of novel treatment regimens to prevent renal damage during cisplatin chemotherapy [7]. 

Designing tools and therapeutic leads from natural product inspiration has long been significant in the chemical sciences. The resurgence of interest in natural products as potential sources for novel drug discovery may be aided by contemporary chemical and bioinformatic methods. In recent times, modern analytical techniques have exponentially contributed to the qualitative and quantitative evaluation of medicinal plants, as these consider the complex matrix of phytoconstituents. Quality-based standardization of medicinal plants using modern analytical techniques helps to obviate quality, safety and efficacy issues for their regulatory aspects Network pharmacology is one of the most precise and robust computational approaches for evaluating the multi-mechanistic and therapeutic effects of medicinal plants based on genomic interaction. 

In India, herbal drugs contribute to the six officially recognized, namely, Ayurveda, Yoga, Naturopathy, Unani, Siddha and Homoeopathy, which are called AYUSH, acknowledged for preventing disease and managing the good health of human beings. The AYUSH system includes thousands of plants with ethnopharmacological relevance for the treatment of urinary disorders, metabolic disorders, immune disorders, etc. [8]. In the AYUSH system, several medicinal plants have been acknowledged for treating urinary or metabolic disorders, and among them, some are part of well-recognized and clinically practiced Indian ayurvedic polyherbal sugar-free syrupy formulation “NEERI KFT” for alleviating acute and chronic kidney ailments. This formulation comprises nineteen herbal decoctions (aqueous extract) such as *Boerhaavia diffusa, Cichorium intybus, Solanum nigrum, Tinospora cordifolia, Nelumbo nucifera, Butea monosperma, Tribulus terrestris, Nelumbo nucifera,* etc., which are claimed traditionally for the amelioration of kidney or urinary malfunction. The composition of NEERI KFT and their traditional therapeutic use as per traditional scripters (Ayurvedic Pharmacopeia of India; Unani Pharmacopoeia of India) have been summarized in Appendix A [9]. However, NEERI KFT has been clinically practiced for decades, yet there is less scientific preclinical evidence reported to date that provides detailed scientific facts about the nephroprotective effect and biomolecular approach of NEERI KFT [9]. Considering the factors, the present study is aimed at evaluating the protective effect of NEERI KFT against cisplatin-induced nephrotoxicity via generating real scientific data against this very famous formulation for CKD in India. In this study, qualitative and quantitative evaluation of phytochemical and network pharmacological study was performed to determine the multitheraputic effect of NEERI KFT followed by in vitro and in vivo assessment for further exploration of NEERI KFT nephroprotective effect. 

## 2. Materials and Methods

### 2.1. Reagents and Chemicals

Folin–Ciocalteu (FC) reagent (2973825), sodium carbonate (A07Z/2006/2512/13) and aluminum chloride (E08Z/0908/2603/13) were procured from S.D Fine chem Pvt. Ltd., Mumbai, India. 2,2-Diphenyl-1-picrylhydrazyl (DPPH) (CAS Number: 1898-66-4) and vitamin C (Vit-C) (CAS Number: 50-81-7) were purchased from Sigma Aldrich Co., St. Louis, MO, USA. MTT {(3-(4, 5-dimethylthiazol-2-yl)-2, 5-diphenyltetrazolium bromide) EZcount^TM^, LOT-0000375488)}; cisplatin, Spectrochem Pvt, Ltd. Mumbai, India (CAS 7681-49-4); Dulbecco’s Modified Eagle Medium (DMEM, REF-AT219-5L); fetal bovine serum (FBS-F2442) HPTLC system (CAMAG, Muttenz, Switzerland); TLC silica gel 60 F254 (Merck KGaA, 64271 Darmstadt, Germany); gallic acid (Lot # SI BL 4700V); caffeic acid (Lot # MKBQ5343V); ferulic acid (CAS 1135-24-6); and Nephro Star/Alpha-Keto Analogue (AKA) purchased from Nephron Star Healthcare Pvt. Ltd, New Delhi, India. All solvents and chemicals used were of analytical grade.

### 2.2. Procurement of Sample

NEERI KFT finished product (NKFTFP) and its aqueous extract (NKFTAQE, with no addition of coloring agents, preservatives, sugar, etc.) were procured as gift samples from Aimil Pharmaceutical (India) Limited., New Delhi, India. The samples were stored in airtight containers at 4 °C for further studies. 

### 2.3. Total Phenolic and Flavonoid Contents and Free-Radical Scavenging Activity of NKFTFP and NKFTAQE

The total phenolic and flavonoid content of the NKFTFP and NKFTAQE was determined by using the Folin–Ciocalteu (FC) and aluminum chloride method with some modifications and expressed as mg gallic acid equivalent/gm of the sample (mg GAE/gm sample) [10]. DPPH free-radical scavenging activity, ABTS (2,2’-azino-bis(3-ethylbenzothiazoline-6-sulfonic acid)), nitric oxide (NO) and iron-chelating assay were performed using a standard protocol with some modifications to determine the anti-oxidant effect of NKFTFP and NKFTAQE (dilutions: 250 and 15.62 µg/mL, respectively) [11,12].

### 2.4. HPTLC Profiling and Quantitative Estimation for Simultaneous Separation of Gallic Acid, Caffeic Acid and Ferulic Acid

HPTLC analysis of NKFTFP and NKFTAQE was performed as per the described protocol with some modifications [13]. In brief, 30 mg/mL of each sample and 1 mg/mL of marker compound stock solutions were prepared. Further, a mix concentration (333.33 µg/mL) of each marker was obtained. Toluene: ethyl acetate: glacial acetic acid (6: 3: 1 *v*/*v*/*v*) was used as a solvent system for the development of the TLC plate. After the development of the TLC plate, densitometric scanning was performed at 260 nm.

### 2.5. Network Pharmacology Analysis

In network pharmacology analysis, the identified metabolites from HPTLC analysis were evaluated for their biological interaction with the different genes involved in the pathophysiology of kidney disease. The study was performed as per our previously reported method on the same genes. A protein–protein interaction (PPI) network and compound–proteins interaction was obtained based on the STRING platform (https://string-db.org/, accessed on 12 March 2022) and Cytoscape (version 3.8.2) software. Gene ontology (GO) analysis through the Metascape Gene Analysis (metascape.org) tool was performed to evaluate multiple physiological roles of each gene in the regulation of kidney and associated disorders after the analysis of the compound–disease common target. The analysis covered all of the nearly functional interactions among the expressed proteins–proteins and compound–proteins [14].

### 2.6. In Vitro Cell Line Studies for the Nephroprotective Effect of NEERI KFT

Human embryonic kidney 293 (HEK 293) cells were obtained from the National Centre for Cell Science (NCCS), Pune, India. In vitro cell line studies were carried out to assess the nephroprotective potential of NKFTFP and NKFTAQE against cisplatin-induced nephrotoxicity and oxidative stress assay on HEK 293 cells. The cells were grown in DMEM with 10% FBS, 1% penicillin and 1% streptomycin supplements. The cultivated cell was separated using trypsin to be subcultured further. In a CO_2_ incubator, the cells were cultivated under the prescribed conditions at 37 °C in a humidified environment with 5% CO_2_ and 95% air. The Nikon Eclipse Ti-S inverted phase-contrast microscope (Nikon, Shonan, Japan) was used to observe cells or even check the confluence level of cultivated cells. The cells were subcultured when the confluence reached 70–80%. Following that, the cytotoxicity test for NKFTFP and NKFTAQE was carried out according to the manufacturer’s procedure with certain changes at various concentration ranges (1–1000 g/mL). 

#### 2.6.1. Determination of Nephroprotective Activity of NEERI KFT

Nephroprotective activity for NKFTFP and NKFTAQE was determined with certain changes to the manufacturer’s procedure. Briefly, the well-cultured cells were treated with 100 L at various concentrations (3.9–500 g/mL) of each sample after the stock solution (1 mg/mL) of each sample was generated in DMEM. The plate was then further incubated for 24 h. After that, the cells were given 100 L of cisplatin (13 g/mL) treatment, and the plate was incubated for an additional 24 h. After the treatment period, the old medium was replaced with 100 L of DMEM without phenol red, and then the plate was further incubated for 3 h with the addition of 10 L of the MTT reagent (5 mg/mL). After incubation, the supernatant was removed and the formed formazan crystals were dissolved in 100 µL of a solubilizing agent. The absorbance was measured at 540 nm using a microplate ELISA reader. The protective effect of NEERI KFT was determined in terms of percentage cell viability and the data obtained were represented statistically. The positive control was used as Vit-C [11].

#### 2.6.2. Cellular Antioxidant Activity (CAA) of NEERI KFT

With few modifications, the CAA of NKFTFP and NKFTAQE was determined using reference protocol [15]. Using oxidation-sensitive Dichloro-dihydro-fluorescein diacetate (DCFH-DA) probes, the intracellular production of ROS was assessed. A 96-well plate of precultured HEK293 cells was treated with 100 L of each sample at varying doses (ranging from 62.5 to 500 g/mL) and 100 L of cisplatin. For 3 h, the dish was incubated. After being exposed to cisplatin (13 g/mL) for 45 min, the cells were then rinsed with 50 L of PBS and treated with 100 L of DCFH-DA (10 mol/L). At 485 nm for the excitation wavelength and 530 nm for the emission wavelength, the fluorescence was measured. Vitamin C was used as a positive control.

### 2.7. In Vivo Studies for the Nephroprotective Effect of NEERI KFT

#### 2.7.1. Experimental Animal

The in vivo experimental studies were carried out on female Wistar albino rats obtained from Central Animal House of Jamia Hamdard, approved by the Institutional Animal Ethics Committee, Jamia Hamdard, New Delhi, with approval number 1568. The animals used for the experimental study were weighed successively with an average weight of 225 ± 17.559 g. All the animals were housed in polypropylene cages and accustomed to standard laboratory conditions with light and dark cycles of 12:12 h; temperature: 23 ± 2 °C; and relative humidity 55 ± 5%. The animals were fed a standard pellets diet and delivered unrestricted access to normal saline (ad libitum) throughout experimentation. The studies were performed under the stringent guiding principle of the Committee for the Purpose of Control and Supervision of Experiments on Animals (CPCSEA) and the Institutional Animal Ethical Committee (IAEC) of Jamia Hamdard, New Delhi, India.

#### 2.7.2. Experimental Groups for Nephroprotective Studies

The studies were conducted as per the protocols with some modifications [7,16,17]. The rats were divided into seven groups each containing six rats. Before treatment, each animal was handled individually and observed for its normal behavior and appearance. The rats were divided into eight groups each containing six rats. Group 1: designated as a normal control group received normal saline and chow diet throughout the study period of fourteen days. Group 2: designated as the toxic control group received normal saline throughout the study period with cisplatin (7 mg/kg/day, i.p.) treatment on the last two days of the study period. Groups 3 and 4: designated as drug-treated groups received a high dose (NKFTFPH; 1112.50 mg/kg/day, p.o.) and low dose (NKFTFPL; 667.50 mg/kg/day, p.o.) of the finished product NEERI KFT, respectively, for fourteen days with cisplatin treatment on the last two days. Groups 5 and 6: designated as drug-treated groups received a high dose (NKFTAQEH; 1112.50 mg/kg/day, p.o.) and a low dose (NKFTAQEL; 667.50 mg/kg/day, p.o.) of the NEERI KFT aqueous extract, respectively, for fourteen days with cisplatin treatment on the last two days. Groups 7 and 8: designated as standard control groups received vitamin C (Vit-C) (10 mg/kg/day, p.o.) and alpha keto-analogs (AKA) (65 mg/kg/day, p.o.), respectively, for fourteen days with cisplatin treatment on the last two days. 

The body weight of all the animals was measured before and post-treatment. Blood samples (via retro-orbital bleeding under light ether anesthesia) and urine samples were collected for the biochemical marker analysis. After blood and urine sample collection, rats were euthanized by cervical dislocation for the collection of tissue samples. The kidney tissues were collected for the assessment of oxidative, inflammatory markers and histopathological studies [18,19]. 

#### 2.7.3. Estimation of Biochemical Markers in Serum and Urine

Biochemical analysis was performed on collected serum and urine samples to assess the biochemical changes. In kidney biomarker analysis, blood urea (BU), creatinine (Cr), albumin (Alb) and uric acid (UA) levels were estimated in serum and urine [7,20].

#### 2.7.4. Assessment of Antioxidant Markers

Using the kidney tissues collected from each animal, homogenates (10% *w*/*v*) were made in trisphosphate buffer (50 mM, pH 7.4) and then centrifuged at 1300 rpm for 10 min at 4 °C. The resulting supernatants were collected and used to assess the malondialdehyde (MDA), nitric oxide (NO), superoxide dismutase (SOD), catalase (CAT), glutathione peroxidase (GPx) and glutathione (GSH), levels using spectrophotometric assays [7,21]. 

#### 2.7.5. Assessment of Inflammatory Markers

Inflammatory biomarkers such as TNF-α, IL-Iβ and CASP-3 were assessed for the supernatant obtained from kidney tissue homogenates. The method for estimation of TNF-α, IL-Iβ and CASP-3 was as per the standard protocol of the kit manufacturer. 

#### 2.7.6. Histopathology

Tissue samples of the kidney were fixed in 10% formalin solution for at least 24 h, and then embedded in paraffin blocks. A rotatory microtome was used to cut the tissue samples at 5 μm thickness, mounted on a glass slide, and stained with hematoxylin and eosin solution. Then, tissue specimens were properly dehydrated, cleared, and cover-slipped. The developed slides of samples were examined under a light microscope (Motic, AE2000TRI) [22].

### 2.8. Statistical Analysis

The data were represented statistically as mean  ±  standard deviation (SD) (*n* = 3/6) using one-way ANOVA followed by the Tukey test was used to compare all the pairs of the column. The statistical significance difference was represented in terms of *p*-value and summary. The comparisons were made between the control and toxic group, while the drug treatment groups were compared against toxic groups. The *p*-values < 0.05 were reflected as statistically significant.

## 3. Results

### 3.1. Total Phenolic, Flavonoids and Free-Radical Scavenging Activity of NKFTFP and NKFTAQE

In the present study, total phenols, flavonoids and free-radical scavenging assays were assessed and all the assays are typically based on the free-radical scavenging capacity of NKFTFP and NKFTAQE. The resulting data reveal that the total phenolic content of NKFTFP and NKFTAQE was found as 214.52 ± 1.01 and 238.56 ± 0.98 mg equivalent to gallic acid/gm sample, whereas the total flavonoid content was found as 73.25 ± 0.93 and 86.64 ± 1.59 mg equivalent to Rutin/gm sample, respectively. In DPPH free-radical scavenging activity, the IC_50_ values of NKFTFP, NKFTAQE and Vit-C for the DPPH assay were found as 218.18 ± 8.73, 179.43 ± 9.31 and 66.02 ± 5.32 μg/mL, respectively. The IC_50_ values for ABTS were found as 268.06 ± 12.31, 24.50 ± 10.99, 72.98 ± 4.03 μg/mL. The IC_50_ values for NO were found as 374.71 ± 14.80, 290.01 ± 8.20, 84.28 ± 4.08 μg/mL. The IC_50_ values for iron-chelating activity were found as 151.31 ± 2.867, 141.46 ± 2.84, 58.39 ± 3.61 μg/mL. The data were represented statistically with mean ± SD, as shown in Figure 1. 

### 3.2. HPTLC Quantitative Estimation for Simultaneous Separation of Gallic Acid, Caffeic Acid and Ferulic Acid

HPTLC quantitative estimation for the simultaneous separation of gallic acid, caffeic acid and ferulic acid in NKFTFP and NKFTAQE was performed, successively. The resulting data reveal numerous numbers of major metabolites in both samples, while validation analysis for simultaneous separation of gallic acid, caffeic acid and ferulic acid was found linear, accurate and robust in the wide range of 150–2400 ng/spot. The limit of detection (LOD) and the limit of quantitation (LOQ) were found as 30.022, 26.454 and 18.22 ng/spot and 90.978, 80.164 and 55.220 ng/spot, respectively. The interday and intraday precision were determined as percentage relative standard deviation or the coefficient of variation, and the results found were expressed in the ranges 1.519–4.772, 1.155–4.237, 1.094–3.075 and 1.536–4.938, 1.165–4.114, 1.103–3.012, respectively. The accuracy of the method developed was determined as the percentage drug recovered by spiking 0, 50, 100 and 150% of the standard to the sample in pre-analyzed samples, which exhibited recovery in the ranges 95.78–103.55%, 97.8–100.62% and 100.72–101.49% for gallic acid, caffeic acid and ferulic acid. Afterward, the drug content in NKFTFP and NKFTAQE was calculated and found to be 0.102 ± 0.038%, 0.064 ± 0.003% for gallic acid; 0.121 ± 0.004%, 0.115 ± 0.005% for caffeic acid; and 0.912 ± 0.033%, 0.956 ± 0.0396% for ferulic acid, respectively. HPTLC plate view at 254 nm and chromatograms of NKFTFP and NKFTAQE are summarized in Figure 2. 

### 3.3. Network Pharmacology Analysis

Network pharmacology analysis was conducted to determine the biological interaction of the identified metabolites in the pathophysiology associated with kidney disease. The results showed that among 54 genes, 22 genes (Appendix A) were found to have the most prominent interaction with the metabolites of NEERI KFT. The outcome of the study showed that the constructed network exhibits a number of nodes: 21, number of edges: 81, an average node degree: 7.71, avg. local clustering coefficient: 0.736, expected number of edges: 28, and PPI enrichment *p*-value: 6.66e-16. The analysis showed that each selected gene exhibited significant interaction and even partially connected to each other. The results showed that gallic acid exhibits a significantly strong interaction with genes such as CASPs, ILs, AGTR1, ACE2, SOD1, etc., while caffeic acid and ferulic acid showed comparatively low interaction with the genes involved in the pathophysiology of the kidney disease. The purple color line or edges in the generated network represent the data collected from the experimental resources. The results of the study are summarized in Figure 3. 

### 3.4. In Vitro Cell Line Studies

The in vitro cell line studies were conducted to evaluate the cytotoxicity, nephroprotective and cellular antioxidant activity of NEERI KFT. In the cytotoxicity study, the resultant data showed a statistically significant (*p* < 0.05) and dose-dependent effect of NKFTFP, NKFTAQE and Vit-C. Based on the experimental outcomes, 500 μg/mL concentration was selected as an effective concentration for each drug, while 13 μg/mL concentration was selected as the cytotoxic concentration of cisplatin. The graphical representation of the cytotoxicity assay of NKFTFP, NKFTAQE and Vit-C are summarized in Figure 4A.

Further, the nephroprotective evaluation for the NKFTFP and NKFTAQE were evaluated and the results revealed 65.42 ± 3.57%, 73.27 ± 3.21% and 80.06 ± 4.33% viability at higher concentrations (500 μg/mL) for NKFTFP, NKFTAQE and Vit-C, respectively, against the nephrotoxicity induced by cisplatin (13 μg/mL) on HEK 293 cells. Further, the statistical observations revealed significant (*p* < 0.05) protection by NKFTFP, NKFTAQE and Vit-C in a dose-dependent manner (3.9–500 µg/mL) against nephrotoxicity induced by cisplatin. The high concentration of each sample retained high cell viability with a suggested statistically significant (*p* < 0.05) difference compared to low-dose treatments. The graphical representation of the nephroprotective assay of NKFTFP, NKFTAQE, Vit-C and cisplatin are summarized in Figure 4B.

### 3.5. Cellular Antioxidant Activity

Using DCFH-DA as a fluorescent probe indicator, CAA was determined to assess the intracellular ROS protection of NKFTFP and NKFTAQE against induced oxidative stress by cisplatin on HEK 293 cells. In contrast, the results reveal that NKFTFP, NKFTAQE and Vit-C have significant (*p* < 0.05) efficacy to quench intracellular ROS production induced by cisplatin. However, NKFTAQE showed higher efficacy than NKFTFP but relatively have no significant difference in their bioactivity. NKFTFP, NKFTAQE and Vit-C at higher concentrations (500 µg/mL) have maximum fluorescent reduction as 50.39 ± 1.34%, 58.62 ± 3.85%, 62.28 ± 4.15%, respectively. Further, the statistical representation was made with respect to their corresponding concentration and the statistical data were found statistically significant (*p* < 0.05). The graphical representation of CAA is shown in Figure 4C.

### 3.6. In Vivo Studies for the Nephroprotective Effect of NEERI KFT

The nephroprotective potential of both samples of NEERI KFT (NKFTFP and NKFTAQE) was evaluated on cisplatin-induced nephrotoxic female rats. 

#### 3.6.1. Body Weight Changes

The body weight of animals from each group was measured before treatment and post-treatment successively. The obtained data were represented statistically to respective relative body weight. The statistical observations reveal that administration of cisplatin showed no significant (*p* > 0.05) changes in relative body weight in the cisplatin-treated group as compared to the control group, while oral administration of NKFTFPL, NKFTAQEL and Vit-C ameliorates body weight similar to the control group, although no significant weight changes were observed between drug and standard treated groups, as shown in Appendix A. 

#### 3.6.2. Estimation of Biochemical Markers in Serum and Urine

Biochemical markers were estimated to assess the biochemical changes in the different groups treated with toxicants and drugs. Blood urea, creatinine, albumin, total bilirubin and uric acid levels were estimated in serum and urine. The results showed that two days of treatment with cisplatin causes significant (*p* < 0.05) changes in biochemical markers in serum and urine. The results of biomarkers analysis in serum and urine are summarized in Table 1 and Table 2.

#### 3.6.3. Assessment of Antioxidant Markers

To evaluate the possible effect of NKFTFP and NKFTAQE on oxidative stress, MDA, NO, SOD, CAT, GPx and GSH were assessed in kidney tissue successively. The data obtained reveal that MDA and NO levels were significantly (*p <* 0.05) increased in the cisplatin-treated group compared to the control group, while treatment with NKFTFP and NKFTAQE were significantly (*p <* 0.05) restored to normal. However, there was no significant difference observed between NKFTFP and NKFTAQE, but NKFTAQE showed a comparatively higher effect than NKFTFP against cisplatin-induced oxidative stress. Additionally, Vit-C and AKA significantly ameliorated antioxidant enzymes such as SOD, CAT, GPx and GSH, which were significantly reduced in the cisplatin-treated group, while treatment with NKFTFP and NKFTAQE significantly (*p <* 0.05) restored to normal the efficacy of enzymes associated with the antioxidant defense system. The graphical representations of experimental outcomes are summarized in Figure 5.

#### 3.6.4. Assessment of Inflammatory Markers

Cisplatin-induced inflammatory cytokines in kidney tissue were measured as per standard protocol. The experimental finding reveals significant (*p* < 0.05) elevation of TNF-α, IL-Iβ and CASP-3 in the cisplatin-treated group compared to the control, while oral administration of NKFTFP, NKFTAQE and Vit-C significantly (*p* < 0.05) reversed the induced level of TNF-α, IL-Iβ and CASP-3 to normal. However, the statistical data do not suggest significant changes between NKFTFP and NKFTAQE, but NKFTAQE has a comparatively high potential to reduce the elevated level of inflammatory cytokines against cisplatin-induced nephrotoxicity. The graphical representations of the experimental outcomes are summarized in Figure 6.

#### 3.6.5. Histopathological Evaluation

The effect of cisplatin-induced nephrotoxicity and its amelioration by NKFTFP and NKFTAQE was assessed by histopathological section evaluation of the kidney. The observations reveal a significant protective effect from NKFTFP and NKFTAQE against distorted renal architecture induced after treatment with cisplatin. However, it can be demonstrated that two days of treatment with cisplatin drastically depletes the shape of Bowman’s capsule, proximal tubule, distal tubule, destructured brush border cells and epithelial cells, as observed through histopathological examination. Similar to that of the normal control group, NKFTFP, NKFTAQE and Vit-C restore the distorted architecture of the kidney and ameliorates the normal function of kidney. The histopathological section analysis of the kidney is summarized in Figure 7.

## 4. Discussion

In light of the study, this is the first study to investigate the phytopharmacological evaluation of sugar-free syrupy compound formulation NEERI KFT. Total phenols, flavonoids and in vitro antioxidant assays for NKFTFP and NKFTAQE were estimated and found to have high contents of phenols and flavonoids and the efficacy to quench free radicals, helping to ameliorate kidney dysfunction by decreasing oxidative and inflammatory stress. Previously, it was reported that the high content of phenols and flavonoids and their efficacy to quench free radicals help to ameliorate kidney dysfunction by decreasing oxidative and inflammatory stress [11,23]. 

HPTLC phytochemical analysis considers several major and minor phytochemicals that were identified, while quantitative analysis reveals gallic acid, caffeic acid and ferulic acid are the major metabolites in both samples of NEERI KFT. The previous finding reveals that gallic acid, caffeic acid and ferulic acid are strong candidates to quench the free radicals induced by oxidative stress and similarly reduce inflammation. Oral administration of gallic acid, caffeic acid and ferulic attenuates increased Cr, BUN, NO, MDA, MPO levels and ameliorates antioxidants enzymes such as SOD, GPx, CAT, GSH levels, and potentiates histopathological protection against nephrotoxicity by improving extensive epithelial cell vacuolization, swelling, desquamation, larger tubular lumens and necrosis [24,25,26]. Furthermore, it is well known from epidemiological studies that the intake of dietary supplements or foods that are enriched in antioxidant agents or constituents such as phenolic or flavonoid compounds reduces the risk of renal dysfunction.

In network pharmacology analysis, gallic acid showed prominent interaction with CAPS3, CASP7, AKT, JUN, CDH11, etc., thus alleviating inflammation, oxidative stress and renal toxicity by downregulation of TNF, IL-1B, AKT, CASP3 and STAT [9,27,28]. Bami and Chowdhury reported the protective effect of ferulic acid and caffeic acid against oxidative and inflammatory stress via regulation of MAPKs CASPs, AGEs, NF-κB, JNK and ERK expression, thus improving the normalcy in kidney function [14,26,29,30,31]. 

However, the effect of NKFTFP and NKFTAQE against cisplatin-induced nephrotoxicity on the HEK 293 cell line was found significant and, in a dose-dependent manner, NKFTAQE showed a comparatively higher effect than NKFTFP. Additionally, cellular antioxidant activity was determined to assess the intracellular ROS protection by NKFTFP and NKFTAQE using DCFH-DA as a fluorescent probe indicator against induced oxidative stress by cisplatin in HEK 293 cells [32]. Owing to enrichment of polyphenols in NEERI KFT, it has a high potential to quench excess free radicals induced by oxidative stress [23,33]. 

In in vivo studies, two days of treatment with cisplatin showed severe weight loss and biomarker modifications in the animals, compared to the control, while the changes were reversed after the oral administration of drugs. In contrast, cisplatin caused a significant elevation in BU, Cr, Alb and UA. Oral administration of NKFTFP and NKFTAQE ameliorates the biochemical changes even at low-dose treatment with the drug, although no significant difference was observed between the drug treatment groups. Additionally, oral administration of Vit-C and AKA as positive control also showed ameliorative effect against elevated levels of biomarkers. Previous preclinical and clinical findings support our experimental study strongly [34,35]. However, increased BU, Cr and UA in serum may be due to an increased catabolic rate of muscles/proteins or amino acids and poor clearance, while increased TP, Alb and Glb may be a reason for increased dehydration, inflammation and increased/altered catabolic ability by cisplatin [36,37]. In a meta-analysis report, the protein level was not significantly affected while Cr, BUN and electrolytes such as P and Ca level significantly decreased in AKA-supplemented individuals [38]. Similarly, with increased disruption of ion channel function, cisplatin-derived cellular stress decreased extracellular Ca, P, K, Cl levels. Notably, previous studies revealed that providing high salt concentrations may competitively prevent cisplatin toxicity [39]. Comparatively, increased levels of ALT, AST, ALP, TB and BD in the cisplatin-treated group can be due to the depletion of hepatocyte and cellular stress, which altered the functional ability of endogenous enzyme heme-oxygenase-1 (HO-1) responsible for bilirubin production. Therefore, BD/TB level in blood was found high due to cisplatin-cellular morbidity [40]. 

Cisplatin affects the detoxification and excretion ability of the kidney vigorously. The impairment of proximal and distal tubules decreases the ability for reabsorption against cisplatin toxicity. Polyuria, proteinuria and reduced Cr clearance may be a result of extensive histological damage and or functional impairment of the kidney due to induced nephrotoxicity [37]. NKFTFP and NKFTAQE significantly improve the ability of kidney function against nephrotoxicity. Similarly, significant elevation of TB and DB levels in urine may represent a sign of favorable or unfavorable effect on the kidney. These findings reveal hyperbilirubinemia’s protective effect against cisplatin-induced nephrotoxicity by reducing oxidative and inflammatory stress. Therefore, it can be suggested that bilirubin is reabsorbed or retained in renal proximal tubular epithelial cells and shows a protective effect against oxidative and inflammatory stress [40]. 

High sodium, calcium and phosphorus levels in urine reflect to the other’s known secondary effects associated with cisplatin toxicity causing excessive urinary loss, predictable changes in intestinal absorption, decreased renal uptake due to the proximal tubular damage and low tissue response of parathyroid hormone and low serum magnesium levels [41]. Further, potassium and chloride levels in urine were found significantly decreased due to the induced intracellular movement of potassium under cellular stress and even retention in the kidney, which may be even reasonable for nephrotoxicity [42]. Similarly, reduced chloride ions in urine represent its competitive association with cisplatin molecules, thus decreasing the sensitivity of the reactive form of cisplatin and resulting in decreased accessibility to DNA or even inducing proximal tubular cell resistance to the apoptotic pathway [35]. Oral administration of NKFTFP and NKFTAQE significantly balances the clearance rate of electrolytes and improves the kidney’s ability to function properly.

Oxidative stress is one of the underlying consequences of cisplatin-induced kidney injury. However, excess production of ROS disturbs physiological function and ultimately damages several organs in the body [43]. Cellular stress due to induced oxidants generally seems to be associated with podocyte injury, excessive proteinuria, progressive focal segmental glomerulosclerosis (FSGS) and sustained tubulointerstitial fibrosis [44]. In our experimental findings, NKFTFP and NKFTAQE significantly improve the efficacy of the antioxidative enzyme system and decrease the level of reactive aldehydes and nitrogen species, while no significant difference was observed in the groups treated with Vit-C and AKA, while both were found potentially active in amelioration of antioxidant enzymes. 

Numerous studies have confirmed that cisplatin-induced oxidative stress and inflammation can activate apoptotic signaling pathways that are involved in the pathogenesis of renal injury. TNF-α, IL-1β and CASP-3 are the proinflammatory mediators, which promote inflammation reaction by stimulating neutrophils and macrophages, ultimately resulting in cellular necrosis or apoptosis [45,46]. In the present study, the experimental outcome suggests a significantly elevated level of cytokines such as TNF-α, IL-1β and CASP-3 in the toxic control group compared to normal control, while oral administration of NKFTFP and NKFTAQE exhibits a significant reduction in oxidative stress and inflammation caused by cisplatin and improves the overall structural and functional ability of the kidney [11].

The histopathological outcomes of the study suggest drastically induced glomerulonephritis, distortion of Bowman’s capsule, proximal tubule, destructured brush border cells and epithelial cells in the cisplatin-treated group compared to normal control. It has been reported that the most common clinical conditions where interstitial hemorrhage occurs are renal infarction, infectious nephritides, ANCA-related vasculitis, trauma, coagulopathies and acute cellular/antibody-mediated rejection in kidney transplant recipients. The kidney erupted with loss of physiological function through elevation and accumulation of toxic end products in blood. Renal injury in rats can be vigorously caused by biochemical changes and homeostasis. Further, decreased affinity of the antioxidant enzymes defense system and autoimmunity protection against induced oxidative and inflammatory stress can be the marked points for renal injury [43,47]. Hence, the present study reveals that NEERI KFT is one of the best alternatives to cure kidney disease and associated disorders, or even can be the best adjuvant to overcome cisplatin toxicity during chemotherapy. 

## 5. Conclusions

The present study concluded that NEERI KFT (NKFTFP and NKFTAQE) exerts multimechanistic and therapeutic action in alleviating acute and chronic kidney dysfunction and their associated pathophysiological complications. Network pharmacology analysis showed significant biological interaction of NEERI KFT metabolites, especially polyphenols, which regulate the expression of several genomes (CASPs, ILs, AGTR1, AKT, ACE2, SOD1, etc.) that are involved in the pathophysiology of kidney disease. In vitro and in vivo investigations showed significant ameliorative effect on kidney histology and biochemical levels via reducing oxidative and inflammatory stress against toxicity induced by cisplatin. Moreover, further molecular and clinical-based investigations are needed to determine the exact molecular approach for its nephroprotective action and to enhance the credibility and therapeutic applicability of India’s famous NEERI KFT formulation. Hence, the present study reveals that NEERI KFT is one of the best alternatives or adjuvants for the treatment of kidney disease, even overcoming drug-induced toxicity during chemotherapy. 

## Figures and Tables

**Figure 1 biomedicines-11-00168-f001:**
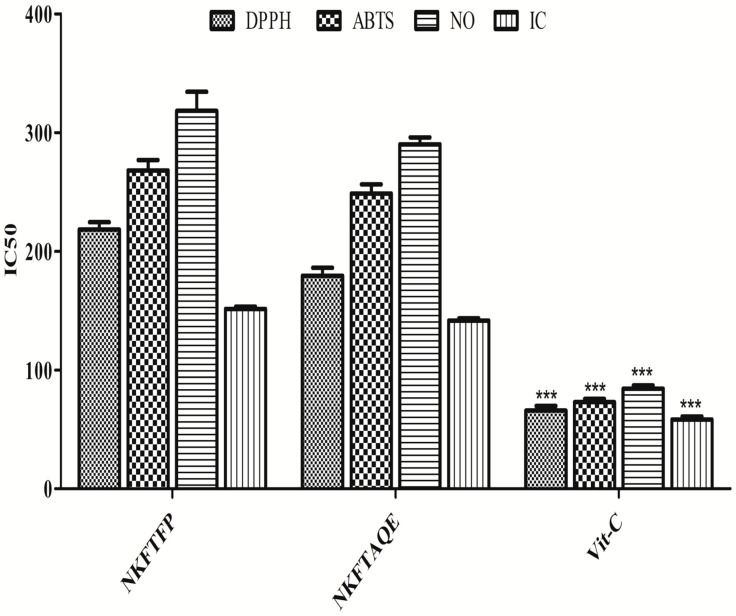
Free-radical scavenging capacity of NKFTFP and NKFTAQE. The statistical representation was made as mean ± SD (*n* = 3). The comparisons are with respect to IC_50_ for each sample. The statistical significance level is expressed at *** *p* < 0.001. The data represented by ‘***’ in Vit-C group showed significant variation in the activity then others.

**Figure 2 biomedicines-11-00168-f002:**
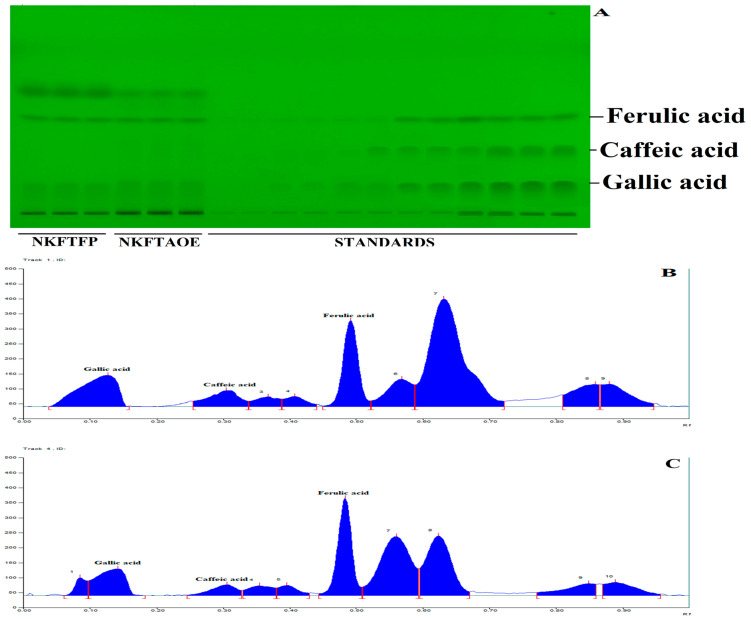
HPTLC profile of NEERI KFT, (**A**) represents the HPTLC developed plate at 254 nm, (**B**,**C**) show HPTLC chromatograms for NKFTFP and NKFTAQE at 260 nm, respectively.

**Figure 3 biomedicines-11-00168-f003:**
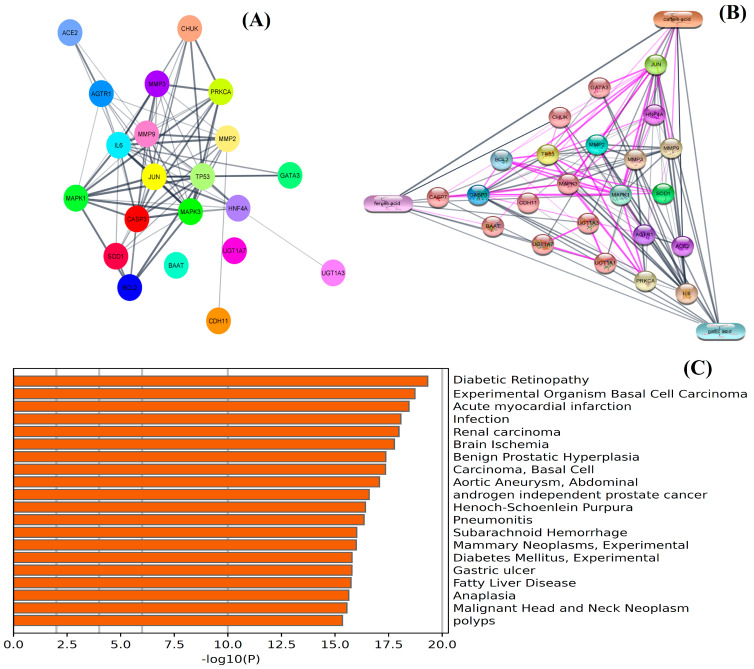
Network pharmacology analysis: (**A**) represents PPI network and (**B**) represents CPI network of integrated genes, while (**C**) represents GO analysis of screened genomes in the pathophysiology of kidney disease and associated disorders.

**Figure 4 biomedicines-11-00168-f004:**
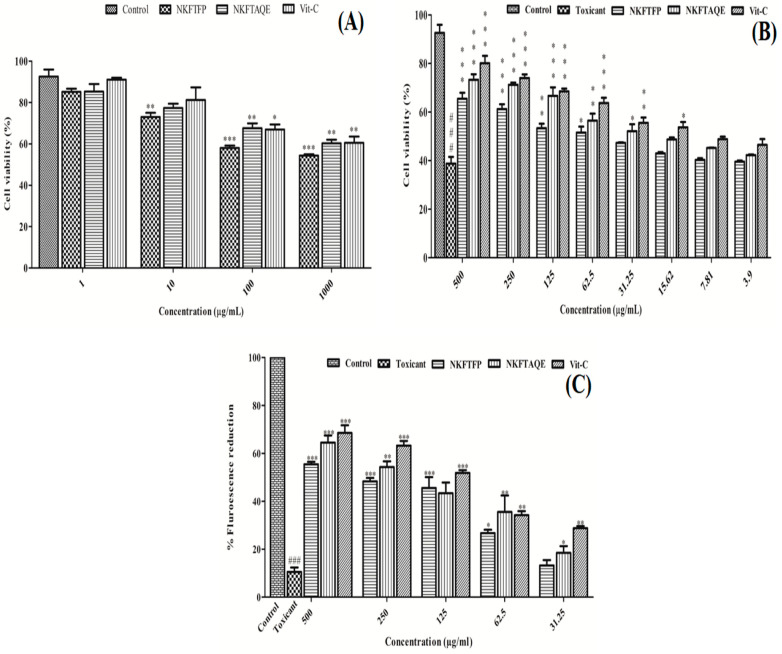
Cytotoxicity (**A**), nephroprotective (**B**) and cellular antioxidant potential (**C**) of NKFTFP and NKFTAQE on HEK 293 cells. The statistical representations are shown as mean ± SD (*n* = 3) using one-way ANOVA followed by the Tukey test. The comparisons are made between the control group to toxicant group, toxicant group to drug-treated group. The statistical significance level is expressed at * *p* < 0.05 (less significant), ** *p* < 0.01 (significant) and ###/*** *p* < 0.001 (high significant). The data represented by the ‘*/**/***’ showed statistical difference in the activity as compared to the toxicant group.

**Figure 5 biomedicines-11-00168-f005:**
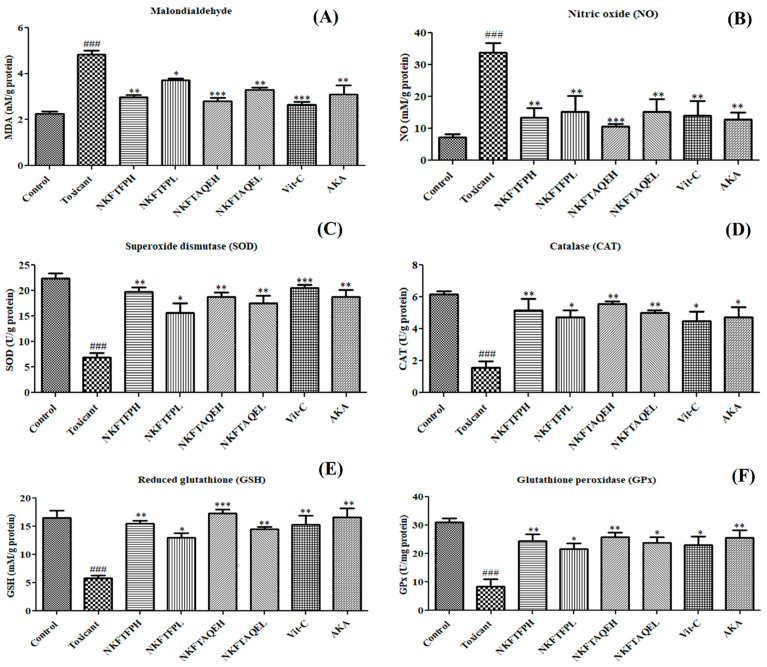
Effect of NKFTFP and NKFTAQE on MOA (**A**), NO (**B**), SOD (**C**), CAT (**D**), GSH (**E**) and GPx (**F**), which were observed against cisplatin-induced oxidative stress in the kidney. The resulting data are expressed statistically as mean ± SD (*n* = 6) using one-way ANOVA followed by the Tukey test to compare all pairs of the column. The comparisons are for control to toxic (###) and toxic to drug-treated group (*/**/***). The statistical significance level is expressed at * *p* < 0.05 (less significant), ** *p* < 0.01 (significant) and ###/*** *p* < 0.001 (high significant). The data represented by ‘*/**/***’ showed statistical difference in the activity as compared to the toxicant group.

**Figure 6 biomedicines-11-00168-f006:**
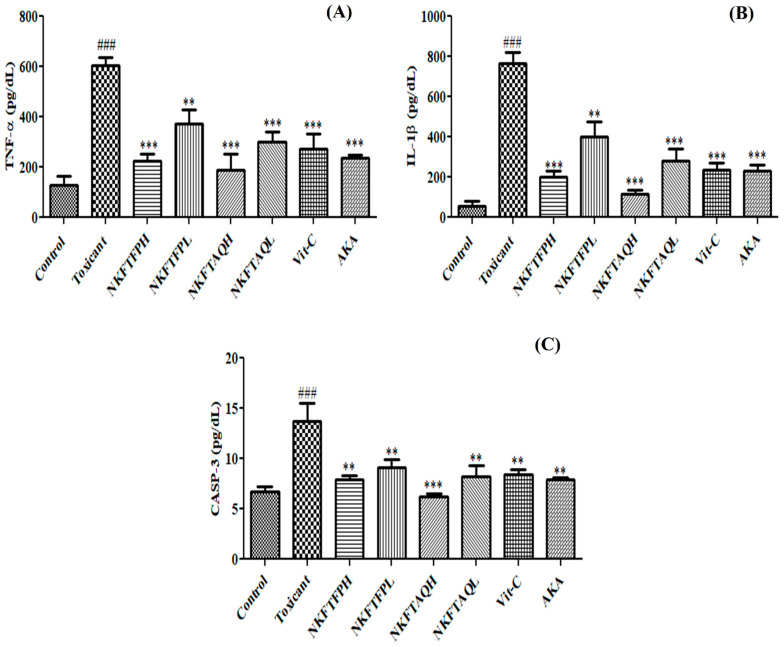
Effect of NKFTFP and NKFTAQE against inflammatory cytokines TNF-α (**A**), IL-1β (**B**) and CASP-3 (**C**) induced by cisplatin in kidney tissues. The resulting data are expressed statistically as mean ± SD (*n* = 6) using one-way ANOVA followed by the Tukey test to compare all pairs of the column. The comparisons are made between control to toxic (###) and toxic to drug-treated group (**/***). The statistical significance level is expressed at ** *p* < 0.01 (significant) and ###/*** *p* < 0.001 (high significant). The data represented by ‘**/***’ showed statistical difference in the activity as compared to the toxicant group.

**Figure 7 biomedicines-11-00168-f007:**
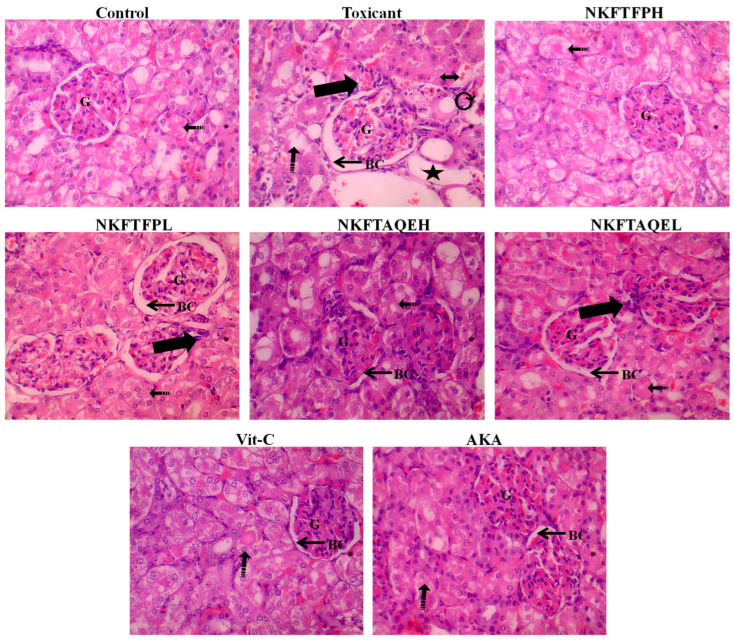
The histopathological section evaluation of the kidney displayed severe inflammatory cell infiltration (thick arrow), irregular dilated Lumina (star), interstitial hemorrhage (dotted arrow), vacuolated cytoplasm and destructured brush border cells (double-sided arrow), atrophy and tubular epithelial injury (curvy arrow), with widened Bowman’s space along with congestion in capillaries of the glomerulus in toxic control group, as compared to the control group.

**Table 1 biomedicines-11-00168-t001:** Assessment of biochemical markers in serum for the experimental groups.

Parameters	Control	Toxicant	NKFTFPH	NKFTFPL	NKFTAQEH	NKFTAQEL	Vit-C	AKA
Blood urea (mg/dL)	22.37 ± 1.732	63.83 ± 2.241 A	30.4 ± 2.771 a	34.89 ± 1.074 a	28.73 ± 1.711 a	31.04 ± 0.438 a	30.85 ± 4.681 a	27.31 ± 2.764 a
Creatinine (mg/dL)	0.51 ± 0.009	1.61 ± 0.037 A	0.58 ± 0.028 a	0.63 ± 0.005 a	0.52 ± 0.019 a	0.55 ± 0.025 a	0.69 ± 0.027 a	0.47 ± 0.086 a
Total Protein (gm/dL)	6.46 ± 0.547	9.83 ± 0.112 B	6.52 ± 0.471 b	6.76 ± 0.209 b	6.59 ± 0.293 b	7.17 ± 0.180 b	7.55 ± 0. 424 b	8.12 ± 1.03 c
Albumin (gm/dL)	2.54 ± 0.341	5.03 ± 0.217 B	3.07 ± 0.141 c	3.10 ± 0.489 c	2.60 ± 0.350 b	3.34 ± 0.460 ns	3.39 ± 0.350 ns	2.93 ± 0.903 c
Globulin (gm/dL)	3.61 ± 0.260	6.04 ± 0.167 B	4.05 ± 0.101 b	4.54 ± 0.405 c	3.79 ± 0.521 b	3.87 ± 0.103 b	3.90 ± 0.350 b	4.39 ± 0.530 c
Total Bilirubin (mg/dL)	0.12 ± 0.021	0.22 ± 0.016 B	0.18 ± 0.012 c	0.20 ± 0.018 ns	0.18 ± 0.025 c	0.19 ± 0.012 ns	0.18 ± 0.016 c	0.18 ± 0.026 ns
Bilirubin (direct) (mg/dL)	0.04 ± 0.008	0.078 ± 0.005 C	0.06 ± 0.006 c	0.07 ± 0.004 ns	0.06 ± 0.006 c	0.06 ± 0.005 ns	0.06 ± 0.003 ns	0.05 ± 0.012 ns
Uric Acid (mg/dL)	2.29 ± 0.246	4.14 ± 0.125 A	2.37 ± 0.133 b	2.87 ± 0.028 c	2.24 ± 0.0165 a	2.42 ± 0.004 b	2.64 ± 0.280 b	2.89 ± 0.545 c
ALT (U/L)	21.54 ± 4.372	53.95 ± 2.828 A	26.92 ± 0.180 a	46.33 ± 3.660 ns	18.13 ± 3.156 a	40.98 ± 1.886 c	35.81 ± 2.073 b	24.49 ± 4.207 a
AST (U/L)	150.05 ± 5.075	261.09 ± 3.477 A	167.93 ± 5.045 a	187.56 ± 5.389 a	166.61 ± 1.059 a	199.57 ± 2.575 b	203.70 ± 16.623 b	178.65 ± 5.338 a
ALP (U/L)	137.57 ± 2.930	182.87 ± 3.503 A	118.77 ± 3.920 a	159.18 ± 2.591 b	109.56 ± 2.981 a	149.83 ± 3.166 a	144.74 ± 3.368 a	127.91 ± 5.593 a
Phosphorus (mg/dL)	5.74 ± 0.075	4.02 ± 0.338 C	6.70 ± 0.185 c	5.19 ± 0.068 ns	6.26 ± 0.396 c	5.73 ± 0.507 ns	6.70 ± 0.454 c	4.79 ± 1.159 ns
Sodium (mmol/L)	136.31 ± 1.506	161.40 ± 2.880 B	139.31 ± 2.711 b	147.99 ± 2.274 ns	142.59 ± 3.615 b	145.99 ± 4.798 c	144.71 ± 3.772 c	136.67 ± 4.398 b
Potassium (mmol/L)	5.09 ± 0.361	3.66 ± 0.362 C	4.69 ± 0.507 ns	4.60 ± 0.296 ns	5.38 ± 0.450 c	5.18 ± 0.246 ns	4.87 ± 0.604 ns	4.98 ± 0.292 ns
Chloride (mmol/L)	106.79 ± 1.760	94.36 ± 1.801 B	104.47 ± 1.113 c	100.16 ± 1.309 ns	106.32 ± 1.261 b	103.46 ± 1.733 c	106.19 ± 2.903 b	107.22 ± 3.49 bb
Calcium (mg/dL)	11.19 ± 0.775	8.18 ± 0.412 B	10.71 ± 0.443 b	9.26 ± 0.165 ns	11.21 ± 0.236 b	9.94 ± 0.151 c	10.41 ± 0.591 c	10.24 ± 0.318 c

Biochemical analyses of kidney and liver in serum are assessed successively and results are represented statistically as mean ± SD (*n* = 6) using one-way ANOVA followed by the Tukey test to compare all pairs of columns. The comparisons are for control to toxicant (A/B/C) and toxicant to control-drug-treated group (a/b/c) (a). The statistical significance level is expressed at C/c *p* < 0.05 (less significant), B/b *p* < 0.01 (significant) and A/a *p* < 0.001 (high significant). The data which is not statistically significant is represented by “ns”.

**Table 2 biomedicines-11-00168-t002:** Assessment of biochemical markers in urine for the experimental groups.

Parameters	Control	Toxicant	NKFTFPH	NKFTFPL	NKFTAQEH	NKFTAQEL	Vit-C	AKA
Urine urea (mg/dL)	61.28 ± 5.714	31.12 ± 2.325 A	53.62 ± 1.922 b	46.06 ± 1.731 c	53.80 ± 2.164 b	49.32 ± 2.892 c	47.96 ± 6.327 c	54.45 ± 2.347 b
Creatinine (mg/dL)	2.02 ± 0.147	0.55 ± 0.175 A	1.38 ± 0.15 b	1.25 ± 0.055 c	1.37 ± 0.039 b	1.35 ± 0.077 b	1.30 ± 0.218 c	1.35 ± 0.233 b
Total Protein (gm/dL)	5.28 ± 0.057	8.87 ± 0.856 B	5.56 ± 0.392 b	6.32 ± 0.621 c	5.72 ± 0.501 b	6.03 ± 0.608 b	5.98 ± 0.671 b	5.38 ± 0.028 b
Albumin (gm/dL)	2.50 ± 0.167	8.66 ± 1.317 A	2.32 ± 0.169 a	3.04 ± 1.328 b	2.43 ± 0.208 a	2.33 ± 0.210 a	2.81 ± 0.365 b	2.48 ± 0.565 a
Globulin (gm/dL)	1.66 ± 0.189	5.28 ± 0.471 A	2.37 ± 0.268 a	2.65 ± 0.395 a	2.05 ± 0.114 a	3.04 ± 0.198 b	2.80 ± 0.208 a	1.77 ± 0.332 a
Total Bilirubin (mg/dL)	0.16 ± 0.010	0.36 ± 0.029 A	0.18 ± 0.019 a	0.21 ± 0.021 a	0.18 ± 0.015 a	0.18 ± 0.024 a	0.20 ± 0.014 a	0.17 ± 0.010 a
Bilirubin (direct) (mg/dL)	0.06 ± 0.005	0.17 ± 0.026 B	0.08 ± 0.021 b	0.10 ± 0.014 c	0.08 ± 0.011 b	0.10 ± 0.011 c	0.09 ± 0.006 b	0.09 ± 0.021 c
Uric Acid (mg/dL)	2.79 ± 0.953	8.14 ± 0.125 A	4.66 ± 0.965 b	4.73 ± 0.142 b	3.68 ± 0.367 b	5.05 ± 0.103 c	5.14 ± 0.987 b	4.48 ± 0.324 b
Phosphorus (mg/dL)	1.38 ± 0.288	4.08 ± 0.410 B	1.92 ± 0.220 c	2.80 ± 0.790 ns	2.27 ± 0.141 c	2.60 ± 0.335 ns	3.15 ± 0.256 ns	3.21 ± 0.509 ns
Sodium (mmol/L)	83.31 ± 2.740	110.37 ± 2.812 A	89.32 ± 2.749 a	96.41 ± 2.764 c	88.79 ± 1.187 a	91.36 ± 2.685 b	92.41 ± 4.151 b	90.54 ± 1.520 b
Potassium (mmol/L)	2.61 ± 0.362	1.13± 0.209 B	2.37 ± 0.290 b	1.33 ± 0.072 ns	2.71 ± 0.178 b	2.15 ± 0.108 c	2.11 ± 0.191 c	2.45 ± 0.254 b
Chloride (mmol/L)	70.81 ± 4.878	50.29 ± 1.321 B	65.82 ± 3.606 c	60.39 ± 1.477 ns	67.80 ± 0.798 c	63.35 ± 5.649 ns	66.35 ± 2.835 c	65.40 ± 2.764 c
Calcium (mg/dL)	3.54 ± 0.574	7.23 ± 0.334 A	4.13 ± 0.062 b	5.66 ± 0.372 ns	4.67 ± 0.566 b	4.99 ± 0.234 b	5.23 ± 0.557 c	4.76 ± 0.318 b

Biochemical analysis of kidney are assessed in urine successively and results are represented statistically as mean ± SD (*n* = 6) using one-way ANOVA followed by the Tukey test to compare all pairs of columns. The comparisons are for control to toxicant (A/B) and toxicant to drug-treated group (a/b/c). The statistical significance level is expressed at c *p* < 0.05 (less significant), B/b *p* < 0.01 (significant) and A/a *p* < 0.001 (high significant). The data which is not statistically significant is represented by “ns”.

## Data Availability

Not applicable.

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
