# Peer review of "Multi-Mechanistic and Therapeutic Exploration of Nephroprotective Effect of Traditional Ayurvedic Polyherbal Formulation Using In Silico, In Vitro and In Vivo Approaches"

_biomedicines, 2023, doi:10.3390/biomedicines11010168_

Round 1

Reviewer 1 Report

This study identifies the main compounds in a traditional Ayurvedic polyherbal preparation, NEERI KFT, and investigates its nephroprotective effects mechanistically in vivo and in-vitro. The manuscript is well written and methods are straightforward and clearly described. For this reason, I only have some minor suggestions to improve the manuscript:

Minor:

1.      Please correct the figure legends, which do not match with their figures.

2.      In result 3.6.5., authors should clarify and describe the changes in detail by histological examination, instead generally describing it as “destructured”.

3.      In figure 7, how did authors identify interstitial hemorrhage? Why is the anatomical structure with hemorrhage indicated by dotted arrow in Toxicant group different from other groups?

4.    In figure 7, I guess BC stands for Bowman’s capsule and G stands for glomerulus. They are wrongly labelled anatomically. 

Author Response

REVIEWER 1

S. No.

Comments

Response

1.       

Please correct the figure legends, which do not match with their figures.

Figure ligands for each figure has been checked and corrected as per suggestion.

2.       

In result 3.6.5., authors should clarify and describe the changes in detail by histological examination, instead generally describing it as “destructured”.

In result 3.6.5., histological examination has been discussed properly for better understanding of the readers.

3.       

In figure 7, how did authors identify interstitial hemorrhage? Why is the anatomical structure with hemorrhage indicated by dotted arrow in Toxicant group different from other groups?

Interstitial hemorrhage in the kidney tissue was examined through the histological examination which has been marked in the histological images. Furthermore, the brief discussion about interstitial hemorrhage has been discussed in the revised manuscript.

4.       

In figure 7, I guess BC stands for Bowman’s capsule and G stands for glomerulus. They are wrongly labelled anatomically.

Figure 7 has been corrected as per suggestions.

Reviewer 2 Report

This study investigated the mechanism underlining Ayurvedic poly herbal formulation treatment. The rational behind the experiment was clear and straight forward. The manuscript is almost well written. The authors should mentioned in the method section of the abstract more details about the cisplatin administration. 

While many different sources are used to set up the study in the introduction, little previous evidence is stated. The introduction is thus short and poorly sets up the rationale for the study. More attention to how this study fits into previous work in natural compounds and inflammation should be added to improve this section.

Please refer to doi: 

10.3389/fvets.2020.00136, 10.1038/nchem.2479

Please describe how the biochemical and antioxidant markers were performed.

There are some minor grammar issues that should be fixed in order to aid the accessibility of the results to

the reader. 

The discussion does a good job at explaining the importance of the results in the context of the inflammatory

pathways involved. However, incorporation of previous results from other related studies is lacking.

Incorporating comparisons with other studies would increase the strength of the paper.

Author Response

Comments and response

Reviewer 2

S. No.

Comments

Response

1.       

This study investigated the mechanism underlining Ayurvedic poly herbal formulation treatment. The rationale behind the experiment was clear and straight forward. The manuscript is almost well written. The authors should mention in the method section of the abstract more details about the cisplatin administration. 

The necessary changes have been done in abstract as per suggestions.

2.       

While many different sources are used to set up the study in the introduction, little previous evidence is stated. The introduction is thus short and poorly sets up the rationale for the study. More attention to how this study fits into previous work in natural compounds and inflammation should be added to improve this section. Please refer to doi: 10.3389/fvets.2020.00136, 10.1038/nchem.2479

This section has been revised as per suggestion followed by citing the provided references.

3.       

Please describe how the biochemical and antioxidant markers were performed.

Biochemical and antioxidant markers estimation were performed as per referenced protocols (Ibrahim et al., 2021; Kpemissi et al., 2019a; Sultana et al., 2012), (Kpemissi et al., 2019b)

4.       

There are some minor grammar issues that should be fixed in order to aid the accessibility of the results to the reader. 

The manuscript has been checked for grammatical and typo-errors, thoroughly.

5.       

The discussion does a good job at explaining the importance of the results in the context of the inflammatory pathways involved. However, incorporation of previous results from other related studies is lacking. Incorporating comparisons with other studies would increase the strength of the paper.

The discussion section has been revised as per suggestions.

References

Ibrahim, M., Parveen, B., Zahiruddin, S., Gautam, G., Parveen, R., Ahmed, M., Arun, K., Sayeed, G., 2021. Analysis of polyphenols in Aegle marmelos leaf and ameliorative efficacy against diabetic mice through restoration of antioxidant and anti- ­ inflammatory status 1–15. https://doi.org/10.1111/jfbc.13852

Kpemissi, M., Eklu-Gadegbeku, K., Veerapur, V.P., Negru, M., Taulescu, M., Chandramohan, V., Hiriyan, J., Banakar, S.M., NV, T., Suhas, D.S., Puneeth, T.A., Vijayakumar, S., Metowogo, K., Aklikokou, K., 2019a. Nephroprotective activity of Combretum micranthum G. Don in cisplatin induced nephrotoxicity in rats: In-vitro, in-vivo and in-silico experiments. Biomed. Pharmacother. https://doi.org/10.1016/j.biopha.2019.108961

Kpemissi, M., Eklu-Gadegbeku, K., Veerapur, V.P., Potârniche, A.V., Adi, K., Vijayakumar, S., Banakar, S.M., Thimmaiah, N. V., Metowogo, K., Aklikokou, K., 2019b. Antioxidant and nephroprotection activities of Combretum micranthum: A phytochemical, in-vitro and ex-vivo studies. Heliyon. https://doi.org/10.1016/j.heliyon.2019.e01365

Sultana, S., Verma, K., Khan, R., 2012. Nephroprotective efficacy of chrysin against cisplatin-induced toxicity via attenuation of oxidative stress. J. Pharm. Pharmacol. https://doi.org/10.1111/j.2042-7158.2012.01470.x
